# A network of autism linked genes stabilizes two pools of synaptic GABA$_A$ receptors

Xia-Jing Tong[1,2†], Zhitao Hu[1,2†‡], Yu Liu[1,2], Dorian Anderson[1,2], Joshua M Kaplan[1,2]*

[1]Department of Molecular Biology, Massachusetts General Hospital, Harvard Medical School, Boston, United States; [2]Department of Neurobiology, Harvard Medical School, Boston, United States

**Abstract** Changing receptor abundance at synapses is an important mechanism for regulating synaptic strength. Synapses contain two pools of receptors, immobilized and diffusing receptors, both of which are confined to post-synaptic elements. Here we show that immobile and diffusing GABA$_A$ receptors are stabilized by distinct synaptic scaffolds at *C. elegans* neuromuscular junctions. Immobilized GABA$_A$ receptors are stabilized by binding to FRM-3/EPB4.1 and LIN-2A/CASK. Diffusing GABA$_A$ receptors are stabilized by the synaptic adhesion molecules Neurexin and Neuroligin. Inhibitory post-synaptic currents are eliminated in double mutants lacking both scaffolds. Neurexin, Neuroligin, and CASK mutations are all linked to Autism Spectrum Disorders (ASD). Our results suggest that these mutations may directly alter inhibitory transmission, which could contribute to the developmental and cognitive deficits observed in ASD.

*For correspondence: kaplan@molbio.mgh.harvard.edu

†These authors contributed equally to this work

Present address: ‡Clem Jones Centre for Ageing Dementia Research, Queensland Brain Institute, The University of Queensland, Brisbane, Australia

Competing interests: The authors declare that no competing interests exist.

## Introduction

Fast synaptic inhibition is primarily mediated by the neurotransmitter GABA and GABA-activated chloride channels (GABA$_A$ receptors). Several studies suggest that an important mechanism for modulating inhibitory transmission is altered abundance of synaptic GABA$_A$ receptors. In mammalian neurons, variation in the amplitude of miniature inhibitory post-synaptic currents (mIPSCs) is caused by corresponding differences in the abundance of GABA$_A$ receptors at synapses (*Nusser et al., 1997*). Long term potentiation of GABAergic transmission is associated with increased mIPSC amplitudes and increased GABA$_A$ abundance at synapses (*Petrini et al., 2014*) while the converse effects are associated with long term depression (*Bannai et al., 2009*).

GABA$_A$ receptors on the cell surface are mobile, undergoing lateral diffusion in the plasma membrane (*Jacob et al., 2005*). Like all synaptic receptors, GABA$_A$ diffusion is significantly reduced at synapses, resulting in accumulation of receptors at the synapse (*Jacob et al., 2005*; *Thomas et al., 2005*; *Bannai et al., 2009*; *Petrini et al., 2014*). Local confinement of receptors at synapses is termed diffusional trapping and is mediated by binding to cytoplasmic scaffolds (*Choquet and Triller, 2013*). The post-synaptic scaffold that immobilizes GABA$_A$ receptors is proposed to consist of a ternary complex of Gephyrin, Neuroligin-2 (NL2), and collybistin (*Jacob et al., 2005*; *Poulopoulos et al., 2009*). Gephyrin binds directly to the large cytoplasmic loop between the third and fourth transmembrane domains (TM3-4 loop) of GABRA1 and 2 subunits (*Tretter et al., 2008*), thereby confining these receptors at synapses (*Jacob et al., 2005*; *Mukherjee et al., 2011*; *Saliba et al., 2012*). Genetic manipulations impairing the Gephyrin/NL2/Collybistin complex invariably decrease but fail to eliminate synaptic GABA$_A$ receptors (*Kneussel et al., 1999*; *Papadopoulos et al., 2007*; *Poulopoulos et al., 2009*). Thus, it is likely that additional proteins are involved in this process.

**eLife digest** Behaviors ranging from movement to memory are dependent on the activity of extensive networks of cells called neurons. Within these networks, neurons communicate across junctions called synapses. The arrival of an electrical signal called an action potential at the 'presynaptic' neuron on one side of the synapse triggers the neuron to release chemical neurotransmitter molecules into the synapse. These molecules then bind to receptors on the 'postsynaptic' cell on the other side of the synapse.

At excitatory synapses, the binding of neurotransmitter to postsynaptic receptors increases the likelihood that the postsynaptic cell will fire its own action potential. By contrast, at inhibitory synapses the binding of neurotransmitters reduces the chances of the postsynaptic cell firing. Most inhibitory synapses use a type of neurotransmitter called GABA, which exerts its effects mainly by binding to a class of receptors called GABA-activated chloride channels (also known as $GABA_A$ receptors).

$GABA_A$ receptors at inhibitory synapses can themselves be divided into two groups: 'mobile' receptors, which can move within the cell membrane that surrounds the postsynaptic cell; and 'immobilized' receptors that form clusters and cannot move. Recent work in mammalian cells identified a protein complex that anchors $GABA_A$ receptors to the cell's internal skeleton to immobilize the receptors. However, there is evidence to suggest that these are not the only proteins that control the location of the receptors.

By studying the inhibitory synapses formed between neurons and body muscles in the roundworm species *Caenorhabditis elegans*, Tong, Hu et al. now show that different groups of proteins maintain the positioning of immobilized and mobile receptors. Specifically, proteins called LIN-2A (a component of the cell's internal skeleton) and FRM-3 (which joins receptors to the cell's skeleton) immobilize $GABA_A$ receptors, whilst the proteins Neuroligin and Neurexin ensure that mobile $GABA_A$ receptors remain within the synapse.

Disturbances to the activity of inhibitory synapses are often seen in autism spectrum disorders, and so too are mutations in the genes that encode the mammalian equivalents of Neuroligin, Neurexin and LIN-2A. The work of Tong, Hu et al. thus suggests a mechanism by which these mutations might contribute to information processing impairments in people with autism. Further research could now investigate if (and how) other genes linked to autism spectrum disorders alter inhibitory synapses.

Within a post-synaptic element, receptors exhibit heterogenous behavior (*Choquet and Triller, 2013*). At both excitatory and inhibitory synapses, super-resolution imaging suggests that a subset of receptors are localized in immobile nanoclusters (~75 nm in diameter) (*Nair et al., 2013*; *Specht et al., 2013*). These immobile receptors undergo dynamic exchange with diffusing receptors that are confined to synapses. These studies highlight several important questions. Do immobile and diffusing receptors both contribute to IPSCs? Current models propose that post-synaptic currents are mediated by immobilized receptors and synaptic plasticity is mediated by the dynamic exchange of receptors between the diffusing and immobile pools (*Choquet and Triller, 2013*). It has not been possible to genetically test these models because mutations that selectively disrupt the two receptor pools are not available. What are the synaptic scaffolds that stabilize immobilized and diffusing $GABA_A$ receptors? What controls the exchange between the two receptor pools?

Here we utilize the *C. elegans* neuromuscular junction (NMJ) as a model to address these questions. We show that immobilized and diffusing $GABA_A$ receptors are stabilized by two distinct post-synaptic scaffolds both of which contain subunits encoded by genes linked to ASD.

## Results

### Inhibitory synapses contain both mobile and immobile UNC-49/$GABA_A$ receptors

*C. elegans* body muscles receive direct inhibitory input from GABAergic motor neurons (*White et al., 1986*). The $GABA_A$ receptors found at these NMJs contain two subunits (UNC-49B and C) both

encoded by the *unc-49* gene (*Bamber et al., 1999*). Mutants lacking UNC-49 receptors have defects in GABA-activated muscle currents, as assessed by recording miniature inhibitory post-synaptic currents (mIPSCs) and muscle currents evoked by an exogenous GABA agonist (muscimol) (*Figure 1A–E*). An mIPSC corresponds to the current evoked by the fusion of a single synaptic vesicle at a GABAergic NMJ and, consequently, measures the function of synaptic UNC-49 receptors. Muscimol activates all surface UNC-49 receptors (including non-synaptic receptors in the nerve cord and muscle cell bodies). The mIPSC (*Figure 1A–C*) and muscimol-evoked current (*Figure 1D,E*) defects of *unc-49* mutants were rescued by a transgene expressing GFP-tagged UNC-49B receptor in body muscles. Muscimol-evoked currents were significantly larger in GFP-UNC-49B transgenic animals (*Figure 1E*), presumably because this transgene is expressed at higher levels than the endogenous *unc-49* gene. These results demonstrate that the GFP-tag (inserted into the TM3-4 loop) did not impair UNC-49B receptor function.

In wild type animals, GFP-UNC-49B fluorescence exhibits a punctate distribution where each punctum is closely apposed to GABAergic motor neuron nerve terminals (labelled with mCherry-tagged RAB-3) (*Figure 1F*), consistent with prior studies (*Thompson-Peer et al., 2012*). To assess the mobility of synaptic UNC-49B receptors, we measured fluorescence recovery after photobleaching

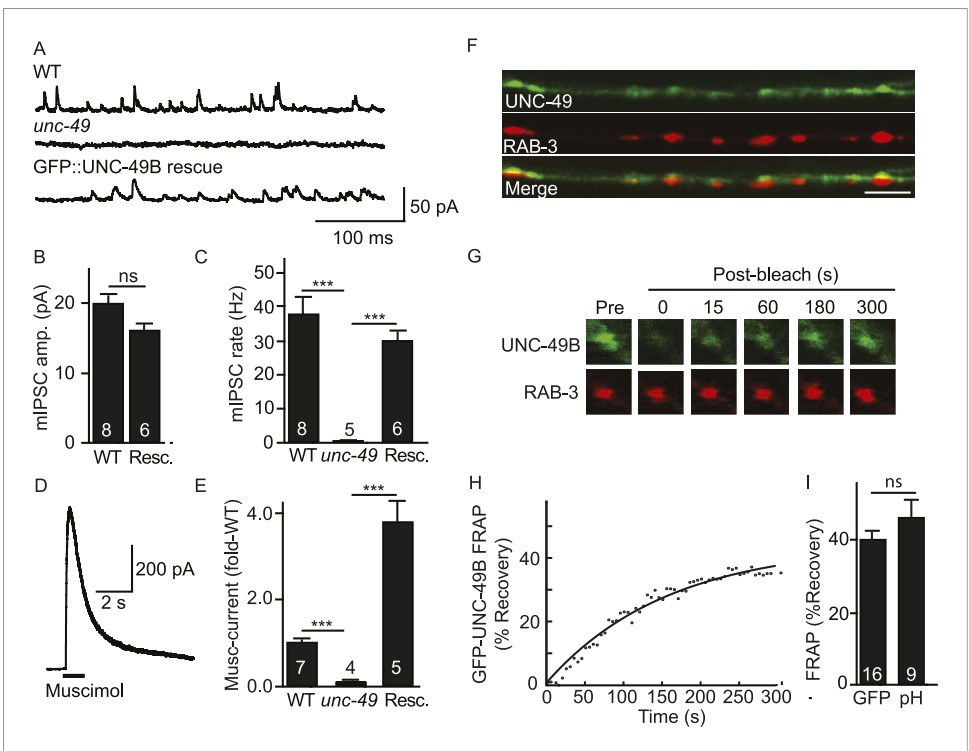

**Figure 1**. Inhibitory NMJs contain both mobile and immobile UNC-49/GABA$_A$ receptors. (**A–E**) mIPSCs and muscimol-evoked currents were abolished in *unc-49* mutants, and were restored by a transgene expressing GFP-tagged UNC-49B in body muscles. mIPSCs (**A–C**) and muscimol-evoked currents (**D**, **E**) were recorded from adult body wall muscles. For mIPSCs, representative traces (**A**), mean current amplitude (**B**) and mean frequency (**C**) are shown. For muscimol-evoked currents, a representative wild type response (**D**), and mean current amplitude (**E**) are shown. GFP-tagged UNC-49B is localized to GABAergic NMJs. (**F**) The distribution of muscle expressed GFP-UNC-49B (Green) is compared to presynaptic RAB-3::mCherry (Red), expressed in GABAergic motor neurons (scale bar 5 µm). (**G–I**) Synaptic UNC-49B consists of both mobile and immobilized receptors. The mobility of synaptic GFP-UNC-49B and pHluorin-tagged UNC-49B (pH-UNC-49B) was analyzed by FRAP. Representative images of GFP-UNC-49B FRAP (**G**), a representative scatter plot of GFP-UNC-49B fluorescence recovery (solid line indicates a single exponential fit) (**H**), and summary data for fluorescence recovery of GFP- and pH-UNC-49B (**I**) are shown. Examples of scatter plots for pH-UNC-49B recovery are shown in *Figure 4—figure supplement 1A*. Pre-synaptic RAB-3::mCherry fluorescence was captured as control. Values that differ significantly are indicated (***, p < 0.001; ns, not significant). The number of animals analysed is indicated for each genotype. Error bars indicate SEM.

(FRAP) of GFP-UNC-49B puncta in the dorsal nerve cord (*Figure 1G–I*). FRAP assesses the mobility of proteins in vivo whereby increased mobility is indicated by increased FRAP. UNC-49B puncta that were co-localized with mCherry-tagged RAB-3 expressed in GABAergic motor neurons were considered synaptic. In wild type controls, 40% of UNC-49B puncta fluorescence was mobile in FRAP experiments, with recovery occurring several minutes after photobleaching (*Figure 1G,H*). GFP-UNC-49B puncta fluorescence could comprise receptors on the cell surface and those in intracellular organelles. To more accurately assess the mobility of surface receptors, we analyzed UNC-49B receptors containing a pH-sensitive GFP (pHluorin) tag in the ecto-domain (pH-UNC-49B). pHluorin fluorescence is quenched in intracellular acidic compartments (e.g., endosomes); consequently, pHluorin fluorescence primarily results from molecules in the plasma membrane (*Miesenbock et al., 1998*). In wild type animals, 46% of pH-UNC-49B synaptic fluorescence recovered following photobleaching (*Figure 1I*). Because similar mobile fractions were observed with the GFP and pHluorin tagged receptors, these results suggest that the majority of UNC-49B puncta fluorescence results from receptors in the plasma membrane. Collectively, these results suggest that UNC-49B synaptic puncta comprise a mixture of mobile (~40% total) and immobilized (~60% total) receptors on the cell surface. Prior studies utilizing both FRAP and single molecule tracking techniques reported similar proportions of mobile and immobilized receptors at both excitatory and inhibitory synapses in cultured mammalian neurons (*Jacob et al., 2005*; *Ashby et al., 2006*; *Heine et al., 2008*).

## FRM-3/EPB4.1 binds the UNC-49B TM3-4 cytoplasmic loop

ERM (Ezrin/Radixin/Moesin) domain containing proteins couple cell surface receptors to the actin cytoskeleton (*Tepass, 2009*) and are implicated in targeting synaptic glutamate receptors and extra-synaptic GABA$_A$ receptors in neurons (*Biederer and Sudhof, 2001*; *Loebrich et al., 2006*). To identify ERM proteins that could play a role in UNC-49B targeting, we screened all *C. elegans* ERM proteins and found that FRM-3 binds the UNC-49B TM3-4 loop in yeast 2-hybrid assays (*Figure 2A*). FRM-3 is a band 4.1 (EPB4.1) paralog. We did three additional experiments to determine if FRM-3 binds UNC-49B in vivo. First, we showed that a *frm-3* promoter construct expressed GFP in body muscles (*Figure 2—figure supplement 1A*), consistent with FRM-3 function in muscles. Second, we showed that GFP-tagged FRM-3 expressed in body muscles formed puncta in the nerve cord that were co-localized with a post-synaptic marker for GABAergic NMJs (mCherry-tagged NLG-1/Neuroligin) (Pearson's correlation R = 0.80 ± 0.028, p = 0.02, n = 8) (*Figure 2B*) (*Maro et al., 2015*; *Tu et al., 2015*). Third, we showed that FLAG-tagged FRM-3 and GFP-UNC-49B co-immunoprecipitated from worm extracts, when both were expressed in body muscles (*Figure 2—figure supplement 1B*). Collectively, these results suggest that FRM-3 is localized to GABAergic synapses where it may directly bind UNC-49B receptors.

## UNC-49/GABA$_A$ synaptic abundance is decreased in *frm-3* EPB4.1 mutants

To estimate the abundance of synaptic GABA$_A$ receptors, we measured the intensity GFP-UNC-49B puncta in the dorsal nerve cord. In *frm-3* mutants, UNC-49B puncta fluorescence was significantly reduced and this defect was rescued by transgenes restoring FRM-3 expression in body muscles but not by those expressed in GABAergic motor neurons (*Figure 2C,D*).

To determine if expression of endogenous UNC-49 receptors was also altered, we patch clamped body muscles and recorded mIPSCs and muscimol-evoked currents. mIPSC rate was unaffected in *frm-3* mutants, implying that pre-synaptic GABA release was not significantly altered (*Figure 2E,G*). mIPSC amplitude was significantly reduced in *frm-3* mutants (*Figure 2E,F*), consistent with decreased UNC-49 abundance at synapses. The *frm-3* mIPSC amplitude defect was rescued by transgenes restoring FRM-3 expression in body muscles (*Figure 2E,F*), implying that FRM-3 acts in muscles to promote the function of synaptic UNC-49 receptors. Muscimol-evoked currents were unaltered in *frm-3* mutants (*Figure 2H*); consequently, the *frm-3* mutant mIPSC defect is unlikely to result from decreased bulk expression and surface delivery of UNC-49 receptors. Collectively, these results suggest that muscle FRM-3 promotes the localization and function of synaptic UNC-49 receptors but is not required for the function or trafficking of non-synaptic UNC-49 receptors.

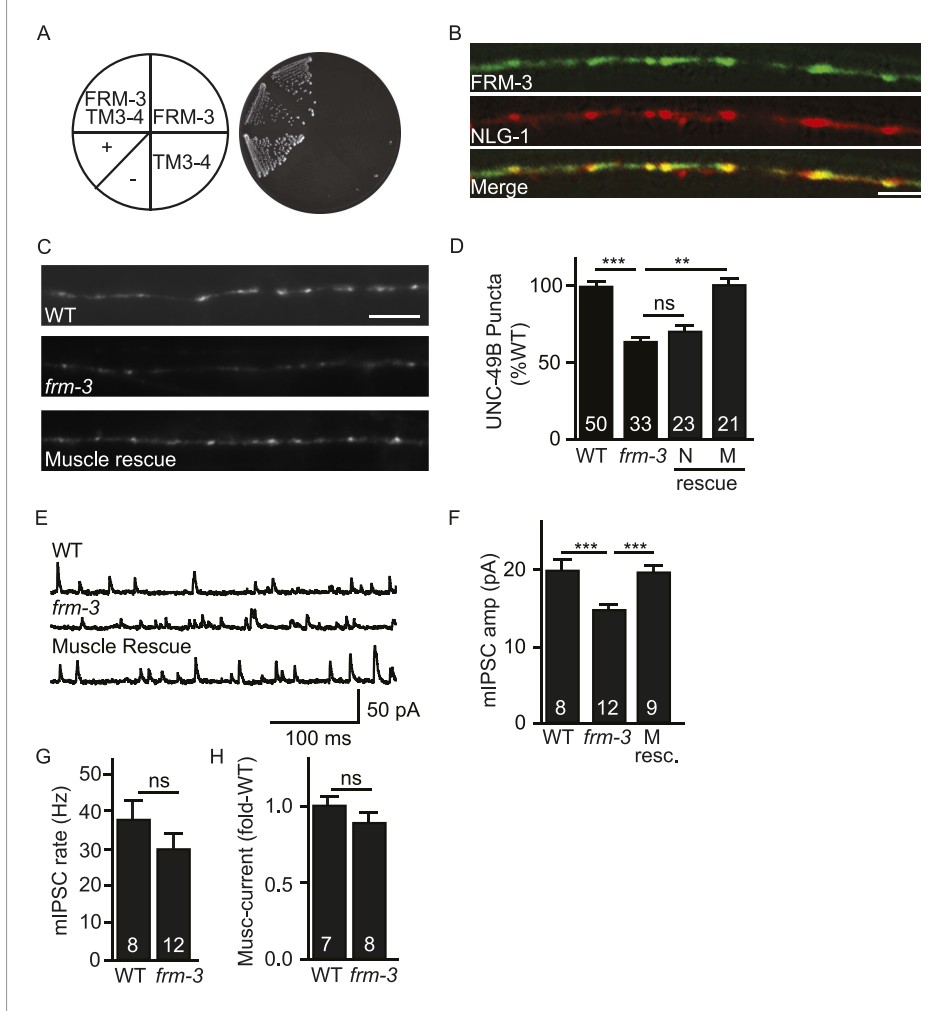

**Figure 2**. FRM-3 EPB4.1 binds UNC-49B and is required for its synaptic targeting. (**A**) FRM-3's ERM domain binds the UNC-49B TM3-4 loop in yeast 2-hybrid assays. Growth of Y2HGold cells on selective media (–Trp/-Leu/-His/-Ade) is shown. Yeast cells were transformed with vectors expressing the indicated fusion proteins. Positive (+, pGBKT7-53 and pGADT7-T) and negative (–,pGBKT7-Lam and pGADT7-T) controls are indicated. (**B**) Muscle expressed FRM-3:: GFP (Green) and NLG-1::mCherry (Red) are co-localized in the nerve cord (scale bar 5 μm). (**C**, **D**) GFP-UNC-49B puncta fluorescence in the nerve cord was decreased in *frm-3* mutants. This defect was rescued by transgenes expressing FRM-3 in body muscles (M) but not by those expressed in GABAergic neurons (N). Representative images (C, scale bar 5 μm) and mean puncta intensity (**D**) are shown. (**E–G**) mIPSC amplitude was decreased in *frm-3* mutants and this defect was rescued by restoring FRM-3 expression in body muscles (M resc.). mIPSCs were recorded from adult body wall muscles. Representative traces (**E**), mean amplitude (**F**) and mean frequency (**G**) are shown. (**H**) The function of total surface UNC-49 receptors was unaltered in *frm-3* mutants. Muscimol-activated currents were recorded from adult body muscles. Mean peak currents are shown. Values that differ significantly are indicated (***, p < 0.001; **, p < 0.01; ns, not significant). The number of animals analysed is indicated for each genotype. Error bars indicate SEM.

The following figure supplement is available for figure 2:

**Figure supplement 1**. FRM-3 is expressed in body muscles and binds to UNC-49B.

## LIN-2A/CASK binds FRM-3 and is also required for synaptic targeting UNC-49B

CASK is a synaptic scaffolding protein that binds directly to EPB4.1 (*Biederer and Sudhof, 2001*). *C. elegans* has two predicted CASK isoforms (LIN-2A and B), both encoded by the *lin-2* gene

(*Hoskins et al., 1996*). LIN-2A and B share the PDZ, SH3, and GK domains while only LIN-2A contains the CaMK homology domain (*Hoskins et al., 1996*). LIN-2A is required for targeting epidermal growth factor receptors (EGFRs) to the basolateral domain of epithelial cells (*Simske et al., 1996*). The impact of LIN-2/CASK on GABA$_A$ receptors has not been determined.

Like their mammalian counterparts (*Biederer and Sudhof, 2001*), LIN-2A/CASK interacted with FRM-3/EPB4.1 in yeast 2-hybrid assays (*Figure 3A*). We did several experiments to determine if LIN-2A acts in muscles to promote UNC-49 targeting to NMJs. A *lin-2* promoter construct expressed GFP in body muscles, indicating that LIN-2A may function in muscles (*Figure 3—figure supplement 1A*). mCherry-tagged LIN-2A expressed in muscles formed puncta in the nerve cords and these LIN-2A puncta were co-localized with GFP-UNC-49B at NMJs, consistent with LIN-2A binding to FRM-3 at NMJs (Pearson's correlation R = 0.68 ± 0.057, p = 0.04, n = 5) (*Figure 3B*). Like *frm-3* mutants, *lin-2* mutants had decreased UNC-49B puncta fluorescence (*Figure 3C,D*) and decreased mIPSC amplitudes (*Figure 3E,F*), both implying that synaptic UNC-49 levels were decreased. mIPSC rates were unaltered in *lin-2* mutants (*Figure 3G*), indicating that pre-synaptic GABA release was unaffected. Muscimol-activated current was unaffected in *lin-2* mutants (*Figure 3H*), indicating that the *lin-2* puncta and mIPSC defects were not caused by decreased bulk expression or surface delivery of UNC-49 receptors. The *lin-2* puncta and mIPSC defects were rescued by transgenes restoring LIN-2A expression in body muscles but not by those expressed in GABAergic motor neurons (*Figure 3D,F*). If LIN-2A and FRM-3 function together to localize UNC-49 receptors, *lin-2* and *frm-3* mutations should not have additive effects in double mutants. Consistent with this idea, UNC-49B puncta fluorescence and mIPSC amplitudes in *frm-3 lin-2* double mutants were not significantly different from those in either single mutant (*Figure 3D,F*). Collectively, these results suggest that LIN-2A/CASK and FRM-3/EPB4.1 function together in body muscles to localize UNC-49B at NMJs but are not required for the expression or function of non-synaptic UNC-49 receptors.

LIN-2/CASK associates with another scaffolding complex that contains LIN-7/Velis and LIN-10/Mint subunits (*Butz et al., 1998*; *Kaech et al., 1998*). We found that mIPSC amplitudes were unaltered in *lin-7* and *lin-10* mutants (*Figure 3—figure supplement 1B,C*), indicating that this complex is not required for UNC-49 synaptic targeting.

## LIN-2A and FRM-3 stabilize immobile UNC-49B receptors at GABAergic NMJs

The decreased UNC-49B synaptic abundance in *lin-2* and *frm-3* mutants could reflect a loss of either mobile or immobilized receptors. To distinguish between these possibilities, we measured FRAP of UNC-49B puncta in the dorsal nerve cord (*Figure 4*). In both *frm-3* (*Figure 4A*) and *lin-2* (*Figure 4B*) mutants, GFP-UNC-49B FRAP was significantly increased (*Figure 4C*). The *frm-3* FRAP defect was rescued by transgenes restoring FRM-3 expression in body muscles (*Figure 4C*). A similar increase in FRAP was observed for pH-UNC-49B receptors in *frm-3* mutants (*Figure 4—figure supplement 1A,B*). The GFP-UNC-49B transgene is likely to be expressed at higher levels than endogenous UNC-49. Over-expression could alter UNC-49B mobility at synapses. To address this possibility, we repeated the FRAP measurements using a single copy RFP-tagged UNC-49B transgene (*krSi2*) (*Pinan-Lucarre et al., 2014*). Using this single copy transgene, a similar increase in FRAP of RFP-UNC-49B was observed in *frm-3* mutants (*Figure 4—figure supplement 1C,D*). Increased FRAP suggests that synaptic UNC-49B receptors had increased ability to undergo exchange in the nerve cord, most likely due to increased diffusional mobility in the plasma membrane. These results suggest that LIN-2A and FRM-3 function as a scaffold that stabilizes an immobile pool of UNC-49 receptors in the plasma membrane at post-synaptic elements.

## NLG-1 neuroligin stabilizes mobile UNC-49B at synapses

Synaptic UNC-49 receptor levels and mIPSC amplitudes were decreased but not eliminated in *frm-3* and *lin-2* mutants; consequently, other proteins must also play a role in confining UNC-49 receptors to these synapses. In mammals, Neuroligin 2 binds gephyrin and collybistin and is required to recruit GABA$_A$ receptors to synapses (*Jacob et al., 2005*; *Poulopoulos et al., 2009*). Prompted by these results, we tested the idea that post-synaptic Neuroligin also plays a role in confining UNC-49 receptors at synapses. Several results support this idea. First, an *nlg-1* promoter construct expressed GFP in body muscles (*Hunter et al., 2010*), consistent with NLG-1 function in muscles. Second,

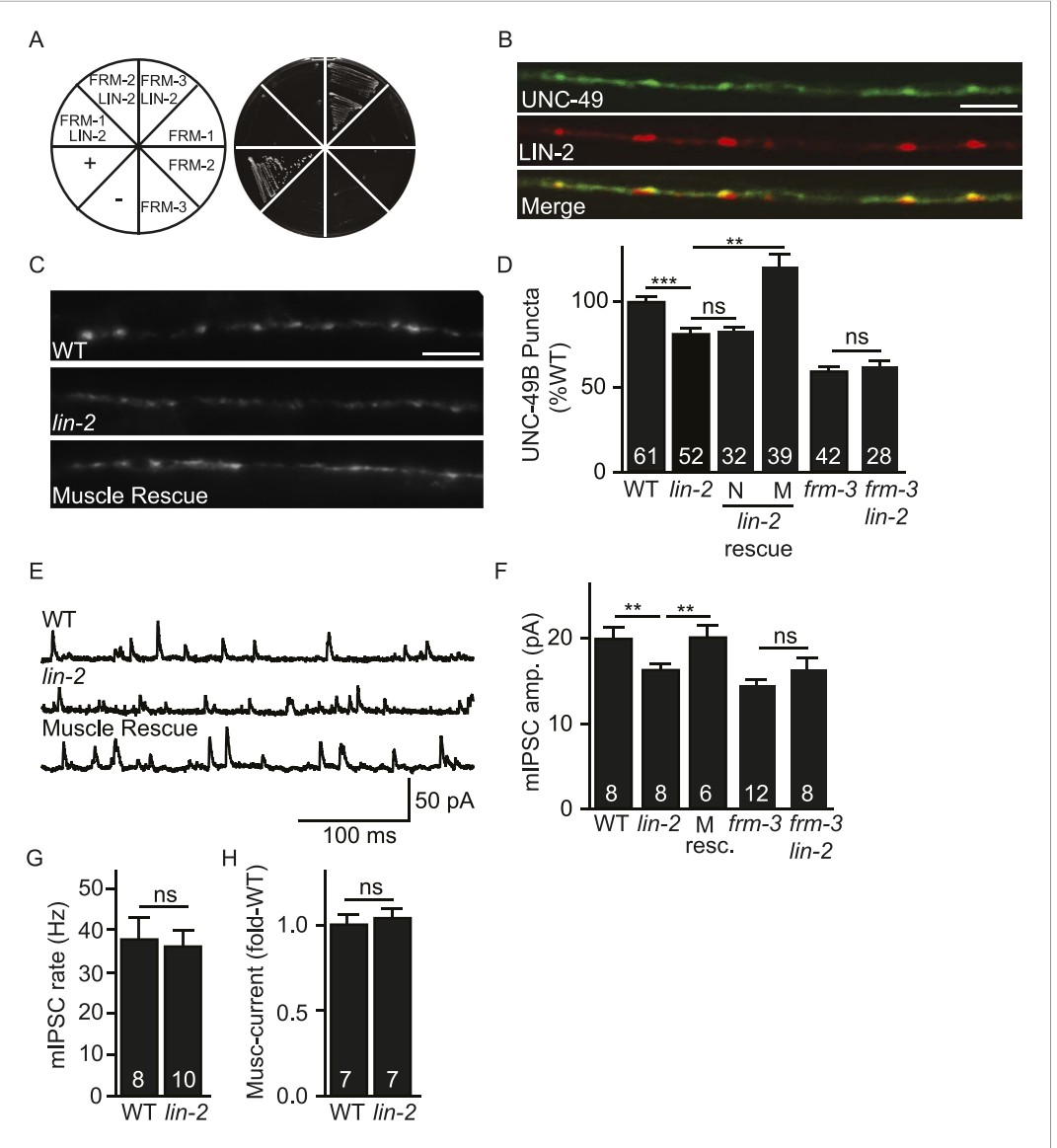

**Figure 3**. LIN-2A/CASK binds FRM-3 and is required for UNC-49 synaptic targeting. (**A**) FRM-3's ERM domain binds LIN-2A in yeast 2-hybrid assays. Growth of Y2HGold cells on selective media (–Trp/-Leu/-His/-Ade) is shown. Yeast cells were transformed with vectors expressing the indicated fusion proteins. Positive (+, pGBKT7-53 and pGADT7-T) and negative (–, pGBKT7-Lam and pGADT7-T) controls are indicated. ERM domains derived from FRM-1, FRM-2 and FRM-3 were tested for interaction with LIN-2A. (**B**) Muscle expressed GFP-UNC-49B (Green) and LIN-2::mCherry (Red) are co-localized in the nerve cord. A representative image is shown (scale bar 5 μm). (**C**, **D**) GFP-UNC-49B puncta fluorescence in the nerve cord was decreased in *lin-2* mutants. This defect was rescued by transgenes expressing LIN-2A in body muscles (M) but not by those expressed in GABergic neurons (N). Representative images (C, scale bar 5 μm) and mean puncta intensity (**D**) are shown. (**E**–**G**) mIPSC amplitude was reduced in *lin-2* mutants and this defect was rescued by a transgene expressing LIN-2 in body muscle (M resc). mIPSCs were recorded from adult body muscles. Representative traces (**E**), mean amplitude (**F**), and mean frequency (**G**) are shown. (**H**) Muscimol-activated currents in adult body muscles were unaffected in *lin-2* mutants, indicating that the function of total surface UNC-49 receptors was unaltered. Mean peak currents are shown. *lin-2* and *frm-3* mutations did not have additive effects on UNC-49B puncta fluorescence (**D**) or mIPSC amplitudes (**F**) in double mutants. Values that differ significantly are indicated (***, $p < 0.001$; **, $p < 0.01$; ns, not significant). The number of animals analysed is indicated for each genotype. Error bars indicate SEM.

The following figure supplement is available for figure 3:

**Figure supplement 1**. mIPSC amplitudes were unaltered in *lin-7* and *lin-10* mutants.

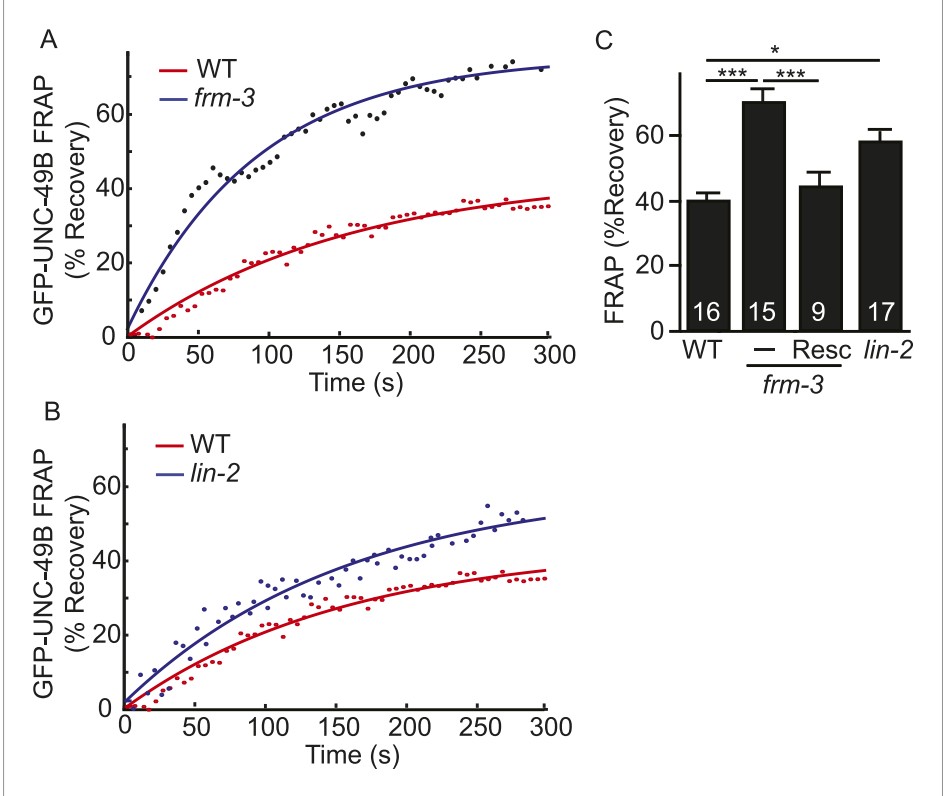

**Figure 4**. LIN-2A and FRM-3 stabilize immobile UNC-49B receptors at synapses. Mobility of synaptic GFP-UNC-49B was analyzed by FRAP. Representative scatter plots of fluorescence recovery (solid lines indicate single exponential fits) (**A**, **B**) and summary data (**C**) are shown. Fluorescence recovery was increased in *frm-3* and *lin-2* mutants, indicating increased mobility of synaptic UNC-49B. The *frm-3* mutant FRAP defect was rescued by a transgene expressing FRM-3 in body muscles (Resc). Values that differ significantly are indicated (***, p < 0.001; *, p < 0.05). The number of animals analyzed is indicated for each genotype. Error bars indicate SEM.

The following figure supplement is available for figure 4:

**Figure supplement 1**. FRAP analysis of pH-UNC-49B and single copy RFP-UNC-49B in *frm-3* mutants.

mCherry-tagged NLG-1 expressed in body muscles was co-localized with GFP-tagged UNC-49B receptors at NMJs (Pearson's correlation R = 0.81 ± 0.02, p = 0.01, n = 8) (*Figure 5A*). Third, GFP-UNC-49B puncta fluorescence was significantly reduced in *nlg-1* mutants (*Figure 5B,C*), consistent with a decrease in total synaptic receptors. Fourth, FRAP of GFP-UNC-49B (*Figure 5D,E*), pH-UNC-49B (*Figure 5—figure supplement 1A,B*), and single copy RFP-UNC-49B (*Figure 5—figure supplement 1C,D*) were all significantly reduced in *nlg-1* mutants, indicating that the residual synaptic UNC-49B receptors were largely immobile. These results suggest that NLG-1 stabilizes a mobile pool of surface UNC-49B receptors at synapses.

To determine if the function of endogenously expressed UNC-49 was altered, we measured GABA activated currents in body muscles. The mIPSC rate was significantly reduced in *nlg-1* mutants (*Figure 5F,G*). The mean mIPSC amplitude (*Figure 5H*) was also significantly reduced, consistent with decreased abundance of synaptic UNC-49. The *nlg-1* mIPSC rate and amplitude defects were both rescued by a transgene restoring NLG-1 expression in body muscles (*Figure 5G,H*). The decreased mIPSC rate in *nlg-1* mutants could be a secondary consequence of the smaller mIPSC amplitudes (i.e. due to decreased detection of mIPSCs). Alternatively, the decreased mIPSC rate could result from decreased pre-synaptic GABA release. The muscimol-evoked current was unaffected in *nlg-1* mutants (*Figure 5I*), indicating that the *nlg-1* puncta and mIPSC defects were not caused by decreased bulk expression or surface delivery of UNC-49 receptors. Collectively, these results suggest that NLG-1

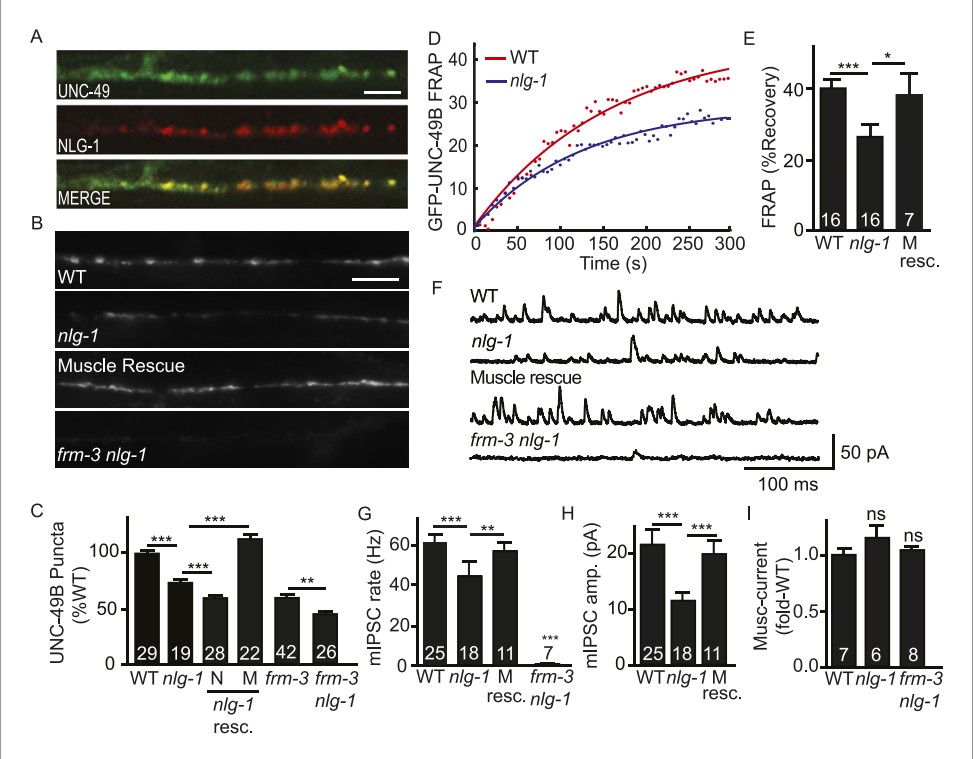

**Figure 5**. NLG-1 stabilizes mobile UNC-49B at synapses. (**A**) Muscle expressed GFP-UNC-49B (Green) and NLG-1::mCherry (Red) are co-localized in the nerve cord. A representative image is shown (scale bar 5 µm). (**B**, **C**) GFP-UNC-49B synaptic abundance was decreased in *nlg-1* mutants and this defect was rescued by transgenes expressing NLG-1 in body muscles (M) but not by those expressed in motor neurons (N). Representative images (B, scale bar 5 µm) and mean puncta intensity (**C**) are shown. (**D**, **E**) Mobility of synaptic GFP-UNC-49B was analyzed by FRAP. Representative scatter plots of fluorescence recovery and single exponential fits (solid lines) (**D**) and summary data (**E**) are shown. Fluorescence recovery was decreased in *nlg-1* mutants, indicating that synaptic UNC-49B mobility was decreased. The *nlg-1* mutant FRAP defect was rescued by a transgene expressing NLG-1 in body muscles (M Resc) (**E**). (**F–H**) mIPSC amplitude was reduced in *nlg-1* mutants and this defect was rescued by a transgene expressing NLG-1 in body muscle (M resc.). mIPSCs were recorded from adult body muscles. Representative traces (**F**), mean frequency (**G**), and mean amplitude (**H**) are shown. (**I**) Muscimol-evoked currents (mean peak amplitude) was unaltered in *nlg-1* and in *frm-3 nlg-1* double mutants. *nlg-1* and *frm-3* mutations had additive effects on UNC-49B puncta fluorescence (**B**, **C**) and mIPSCs (**F**) in double mutants. The number of animals analyzed is indicated for each genotype. Error bars indicate SEM. Values that differ significantly are indicated (***, p < 0.001; **, p < 0.01; *, p < 0.05; ns, not significant).

The following figure supplement is available for figure 5:

**Figure supplement 1**. FRAP analysis of pH-UNC-49B and single copy RFP-UNC-49B in *nlg-1* mutants.

stabilizes a mobile pool of UNC-49 receptors at synapses, and that this receptor pool contributes to post-synaptic currents. These results are consistent with two recent studies, which also showed that NLG-1 promotes UNC-49 targeting to synapses (*Maro et al., 2015*; *Tu et al., 2015*).

## Presynaptic NRX-1α inhibits immobilization of mobile UNC-49B

*C. elegans* expresses long (NRX-1α) and short (NRX-1β) Neurexin isoforms, both encoded by the *nrx-1* gene. NLG-1 binds to the sixth LNS repeat of NRX-1 (*Hu et al., 2012*) and, consequently, could bind to both NRX-1α and β. To test the impact of NRX-1 on UNC-49B localization, we isolated an *nrx-1* null allele (*nu485*) that inactivates both NRX-1α and β. Mean mIPSC amplitude was significantly increased in *nrx-1* null mutants (*Figure 6A,B*), indicating an increased number of functional UNC-49 receptors at synapses. The mIPSC rate was unaltered in *nrx-1* mutants (*Figure 6C*), suggesting that presynaptic

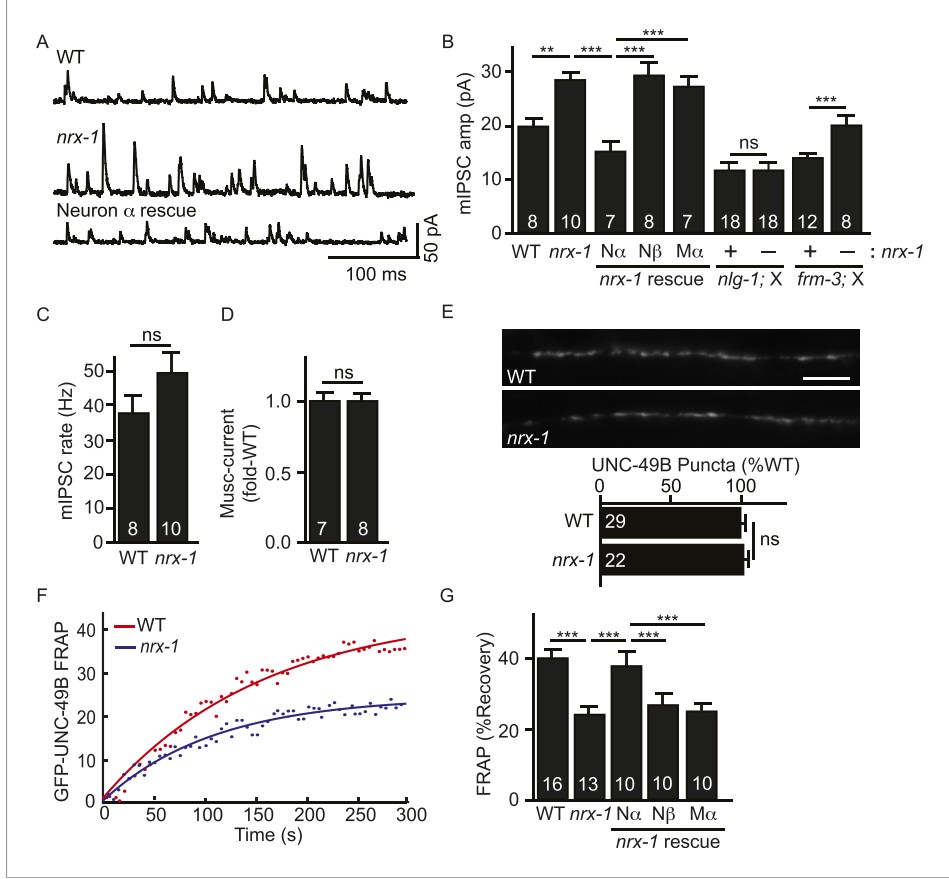

**Figure 6**. Pre-synaptic NRX-1α inhibits immobilization of synaptic UNC-49B. (**A**, **B**) Mutations inactivating *nrx-1* increased mIPSC amplitude and this defect was rescued by transgenes expressing NRX-1α in motor neurons (Nα) but not those expressing NRX-1β (Nβ). Transgenes expressing NRX-1α in body muscles (Mα) lacked rescuing activity. mIPSCs were recorded from adult body muscles. Representative traces (**A**), mean amplitude (**B**) and mean frequency (**C**) are shown. The effect of *nrx-1* mutations on mIPSC amplitudes was eliminated in *nrx-1;nlg-1* double mutants but was unaffected in *nrx-1; frm-3* double mutants (**B**). (**D**) Muscimol-evoked currents (mean peak amplitude) was unaffected in *nrx-1* mutants. (**E**) GFP-UNC-49B synaptic abundance was unaltered in *nrx-1* mutants Representative images (top, scale bar 5 μm) and mean puncta intensity (below) are shown. (**F**, **G**) FRAP analysis suggests that mobility of synaptic GFP-UNC-49B was decreased in *nrx-1* mutants. This FRAP defect was rescued by transgenes expressing NRX-1α in motor neurons (Nα) but not those expressing NRX-1β (Nβ). Transgenes expressing NRX-1α in body muscles (Mα) lacked rescuing activity. Representative scatter plots of fluorescence recovery and single exponential fits (solid lines) (**F**) and summary data (**G**) are shown. The number of animals analyzed is indicated for each genotype. Error bars indicate SEM. Values that differ significantly are indicated (***, $p < 0.001$; **, $p < 0.01$; ns, not significant).

GABA release occurs normally. The amplitude of muscimol-activated current was also unaltered in *nrx-1* mutants (**Figure 6D**); consequently, the increased mIPSC amplitude was unlikely to be caused by increased bulk expression or surface delivery of UNC-49 receptors. GFP-UNC-49B puncta fluorescence was also unaltered in *nrx-1* null mutants (**Figure 6E**), suggesting that the mIPSC amplitude increase was not caused by increased abundance of synaptic UNC-49B receptors. Although UNC-49B puncta fluorescence was unaltered, FRAP of synaptic GFP-UNC-49B was significantly reduced in *nrx-1* mutants (**Figure 6F,G**). The *nrx-1* mutant defects in mIPSC amplitude (**Figure 6B**) and UNC-49B FRAP (**Figure 6G**) were both rescued by transgenes expressing NRX-1α in GABAergic motor neurons but not by those expressing NRX-1β. NRX-1α transgenes expressed in body muscles lacked rescuing activity (**Figure 6B,G**). Collectively, these results suggest that the total number of synaptic UNC-49B receptors was unaltered in *nrx-1* mutants; however, there was a shift in receptor mobility whereby the pool of immobilized synaptic UNC-49B was enlarged (resulting in increased

mIPSC amplitude) while the mobile pool was diminished. These results support the idea that pre-synaptic NRX-1α inhibits the immobilization of mobile UNC-49B receptors at synapses.

Which pool of UNC-49 receptors is required for the increased mIPSC amplitudes in *nrx-1* mutants? To address this question, we recorded mIPSCs in double mutants. We found that the effect of *nrx-1* mutations on mIPSC amplitude was eliminated in *nrx-1; nlg-1* double mutants (*Figure 6B*). By contrast, the *nrx-1* mIPSC amplitude defect was not blocked in *nrx-1; frm-3* double mutants (*Figure 6B*). These results suggest that presynaptic NRX-1α inhibits diffusional trapping of NLG-1-stabilized UNC-49 receptors but has little effect on the mobility of FRM-3-associated receptors.

## FRM-3- and NLG-1-stabilized UNC-49B receptors both contribute to synaptic responses

The preceding results suggest that UNC-49B synaptic puncta comprise two pools of receptors that are stabilized by different scaffolds. Immobilized UNC-49B receptors fail to undergo diffusional exchange in FRAP experiments and are stabilized by FRM-3 and LIN-2A. Mobile UNC-49B receptors mediate fluorescence recovery in FRAP experiments and are stabilized by NLG-1 and NRX-1α. In this scenario, we expect that double mutants lacking both scaffolds would have additive defects, lacking both receptor pools. Consistent with this idea, UNC-49B puncta fluorescence was significantly reduced (*Figure 5B,C*) while GFP-UNC-49B FRAP was significantly increased (*Figure 7A,B*) in *frm-3 nlg-1* double mutants compared to the corresponding single mutants. In *frm-3 nlg-1* double mutants, both the immobile pool of synaptic UNC-49B (*Figure 7A,B*) and mIPSCs (*Figure 5F*) were completely eliminated. Given the absence of mIPSCs, we could not measure quantal size in double mutants. As an alternative, we measured mIPSC rate and found that it was dramatically reduced in *frm-3 nlg-1* double mutants compared to either single mutant (*Figure 5G*). Muscimol-activated muscle current in *frm-3 nlg-1* double mutants did not significantly differ from wild type controls (*Figure 5I*), suggesting that the puncta, FRAP, and mIPSC defects were not caused by decreased bulk expression or surface delivery of UNC-49 receptors. These results support the idea that FRM-3 and NLG-1 stabilized UNC-49B receptors represent two distinct pools of synaptic receptors, which together account for all synaptic UNC-49 receptors.

## Both scaffolds contribute to quantal size variation

Analysis of excitatory and inhibitory synapses in mammals suggests that differences in synaptic receptor abundance contribute significantly to variation in quantal sizes (*Nusser et al., 1997*, *1998*). The coefficient of variation (CV) of the mIPSC amplitudes was ~0.6–7 in wild type animals (*Figure 7C,D*), which is similar to values reported for some mammalian CNS synapses (*Bekkers et al., 1990*; *Hanse and Gustafsson, 2001*). If quantal variability is dominated by diffusion of UNC-49 receptors in and out of the postsynaptic site, the mIPSC amplitude CV should scale inversely with the square root of receptor number. In this scenario, we expect that mutations decreasing UNC-49B punta fluorescence (e.g., *frm-3*, *lin-2*, and *nlg-1*) would produce corresponding increases in the CV of mIPSC amplitudes. Contrary to this idea, CV was significantly reduced in *frm-3* and *nlg-1* mutants and was not significantly altered in *lin-2* mutants (*Figure 7C,D*). The *frm-3* and *nlg-1* mIPSC amplitude CV defects were rescued by transgenes that restore expression of the corresponding genes in body muscles (*Figure 7C,D*). Changes in CV were also not correlated with changes in mean mIPSC amplitude. Mean mIPSC amplitude was increased in *nrx-1* mutants (*Figure 6B*) and decreased in *lin-2* mutants (*Figure 3F*) while CV was unaltered in both cases (*Figure 7C*). Collectively, these results suggest that the FRM-3 and NLG-1 scaffolds increase the diversity of quantal responses, and that this effect cannot be explained by changes in synaptic UNC-49 levels nor by changes in mean mIPSC amplitude.

## Discussion

We have analyzed mechanisms that spatially confine UNC-49/GABA$_A$ receptors at the *C. elegans* NMJ. Our results lead to seven conclusions. First, UNC-49 synaptic puncta comprise both mobile and immobilized receptors, consistent with prior studies analysing mammalian synapses. Second, immobilized UNC-49B is stabilized at synapses via binding to FRM-3/EPB4.1 and LIN-2A/CASK. Third, mobile UNC-49B is stabilized at synapses by post-synaptic NLG-1/Neuroligin. Fourth, pre-synaptic NRX-1α/Neurexin inhibits the immobilization of NLG-1 stabilized UNC-49B. Fifth, the FRM-3 and NLG-1 stabilized receptor pools both contribute to inhibitory synaptic transmission. Sixth, both

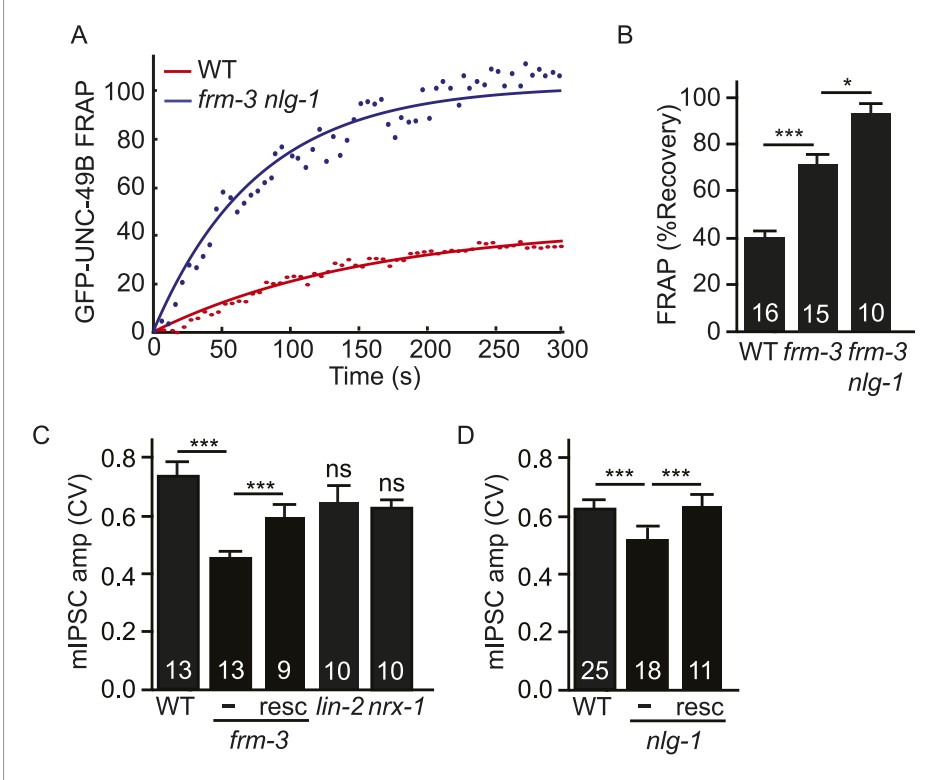

**Figure 7**. Both UNC-49 receptor pools contribute to post-synaptic responses. (**A**, **B**) FRAP analysis suggests that the immobile pool of synaptic GFP-UNC-49B was eliminated in *frm-3 nlg-1* double mutants. Representative scatter plots of fluorescence recovery and single exponential fits (solid lines) (**A**) and summary data (**B**) are shown. (**C**, **D**) The FRM-3 and NLG-1 scaffolds increase the diversity of quantal responses. CV of mIPSC amplitudes are shown for the indicated genotypes. CV was significantly decreased in *frm-3* (**C**) and in *nlg-1* (**D**) mutants. The *frm-3* and *nlg-1* CV defects were rescued by transgenes restoring expression of the corresponding genes in body muscles (resc). CV was not significantly altered in *lin-2* and *nrx-1* mutants (**C**). The number of animals analyzed is indicated for each genotype. Error bars indicate SEM. Values that differ significantly are indicated (***, p < 0.001; *, p < 0.05; ns, not significant).

scaffolds increase the variability of quantal responses. And seventh, all synaptic UNC-49 receptors can be explained by these two scaffolds. Below we discuss the significance of these findings.

## Two pools of synaptic GABA$_A$ receptors

Synaptic UNC-49/GABA$_A$ receptors are confined by two scaffolds that function in parallel. The FRM-3/EPB4.1 and LIN-2A/CASK scaffold accounts for ~40% of synaptic receptors while the NLG-1 scaffold accounts for ~30% of synaptic UNC-49 (assessed by puncta fluorescence). These scaffolds define two pools of synaptic GABA$_A$ receptors that function independently. In double mutants lacking both pools, immobilized synaptic UNC-49 receptors and mIPSCs are eliminated. Mobile receptors detected by FRAP in *frm-3 nlg-1* double mutants likely correspond to extra-synaptic receptors on the cell surface (which mediate muscimol-activated currents). Thus, the FRM-3 and NLG-1 scaffolds together account for all GABA$_A$ receptors at the body wall NMJ. Consequently, the dynamics of GABA$_A$ levels at this synapse will be determined by the detailed biochemistry of these two scaffolds. Our results also suggest that synaptic receptors represent a small fraction of total surface UNC-49 receptors. Mutants lacking both scaffolds have no defect in total surface receptors (assessed by muscimol-activated currents). Thus, these scaffolds have no effect on the assembly, anterograde trafficking, and surface delivery of UNC-49 receptors. Instead, these scaffolds play a specific role in concentrating UNC-49 surface receptors at synapses.

Our results strongly support the idea that both GABA$_A$ pools are required for inhibitory synaptic transmission. Mutations inactivating the FRM-3/LIN-2A scaffold caused similar decreases in immobilized UNC-49B (40% decrease) and mIPSC amplitude (30% decrease). Inactivating NLG-1 caused corresponding decreases in total synaptic UNC-49B (30% decrease), mobile UNC-49B (40% decrease), and mIPSC amplitudes (48% decrease). Thus, analysis of single mutants suggests that FRM-3 and NLG-1 stabilized receptors contribute equally to post-synaptic currents.

## FRM-3/EPB4.1 and LIN-2A/CASK define an immobile pool of synaptic GABA$_A$ receptors

We show that FRM-3/EPB4.1 and LIN-2A/CASK together comprise a scaffold that immobilizes synaptic GABA$_A$ receptors. Other ERM domain containing proteins were previously implicated in neurotransmitter receptor localization. The ERM protein Radixin stabilizes extra-synaptic GABRA5 receptors in mammalian neurons (*Loebrich et al., 2006*), suggesting that different populations of GABA receptors are stabilized by distinct ERM proteins. Mammalian EPB4.1 and its *Drosophila* counterpart (coracle) were previously implicated in synaptic localization of glutamate receptors (*Shen et al., 2000*; *Chen et al., 2005*). Excitatory and inhibitory synaptic transmission were not significantly altered in mouse CASK knockouts (*Atasoy et al., 2007*); however, subtle defects (e.g., the ~20% decrease in quantal size reported here) could have been missed in this study. *Drosophila* CASK mutants have decreased targeting of glutamate receptors to larval NMJs (*Chen and Featherstone, 2011*). A role for CASK in synaptic targeting of GABA$_A$ receptors has not been described.

We found that *frm-3* EPB4.1 and *lin-2* CASK mutants have decreased synaptic GABA$_A$ levels, decreased immobilization of synaptic GABA$_A$ receptors, and corresponding decreases in mIPSC amplitudes. Both mutants had significant residual levels of immobile synaptic GABA$_A$ receptors and post-synaptic currents. Thus, FRM-3 and LIN-2 define a sub-population of immobilized synaptic GABA$_A$ receptors, accounting for ~40% of total synaptic receptors. For all synaptic phenotypes, *lin-2* defects were less severe than those observed in *frm-3* mutants, perhaps because LIN-2/CASK regulates UNC-49 targeting by modulating FRM-3 function. For example, prior studies suggest that ERM proteins equilibrate between active (open) and inactive (closed) conformations (*Pearson et al., 2000*). It is possible that LIN-2/CASK binding stabilizes the active conformation of FRM-3. Collectively, these results suggest that ERM and CASK proteins play a conserved role in targeting neurotransmitter receptors. In each case, CASK and ERM proteins account for a subset of receptors confined to a synapse.

## The UNC-49 pool stabilized by pre-synaptic NRX-1α and post-synaptic NLG-1 consists of both immobilized and diffusing GABA$_A$ receptors

In mammals, Neuroligin 2 works in conjunction with gephyrin and collybistin to localize GABA$_A$ receptors to synapses. Knockout mutations in each of these genes reduce but do not eliminate synaptic GABA$_A$ receptors (*Mukherjee et al., 2011*; *Nair et al., 2013*; *Specht et al., 2013*). Gephyrin binding to the TM3-4 loop has been shown to immobilize GABA$_A$ receptors at synapses (*Jacob et al., 2005*; *Mukherjee et al., 2011*). The impact of Neuroligin 2 mutations on GABA$_A$ receptor mobility has not been determined. Our results are largely consistent with these prior studies. Mutations inactivating NLG-1 decreased but did not eliminate synaptic UNC-49 and caused a corresponding decrease in mIPSC amplitudes. Two recent studies reported similar effects of *nlg-1* mutations on synaptic UNC-49 levels (*Maro et al., 2015*; *Tu et al., 2015*). Thus, as in mammalian neurons, NLG-1 represents one of multiple mechanisms for confining GABA$_A$ receptors at synapses. NLG-1 association with UNC-49 could be mediated by binding to an intermediary Gephyrin-like molecule. The *C. elegans* ortholog of gephyrin is MOC-1; however, its role in synaptic targeting of UNC-49 has not been determined.

Our studies provide further insights into how Neurexin and Neuroligin function to target synaptic GABA$_A$ receptors. Two results suggest that the NLG-1 stabilized pool consists of both mobile and immobilized UNC-49 receptors. First, in double mutants lacking both scaffolds (i.e. *frm-3 nlg-1* double mutants), immobilized UNC-49B receptors and mIPSCs were both eliminated. This result implies that immobile UNC-49B receptors observed in *frm-3* single mutants are derived from the NLG-1-stabilized pool and that immobilization of these receptors does not require FRM-3. Second, mIPSC amplitudes and immobile synaptic UNC-49B levels are significantly increased in mutants lacking presynaptic

NRX-1α. Both of these effects were abolished in *nrx-1; nlg-1* double mutants. These results suggest that pre-synaptic NRX-1α inhibits immobilization of diffusing NLG-1-stabilized UNC-49 receptors. Trans-synaptic NRX-1α/NLG-1 complexes may confine mobile synaptic receptors via low affinity binding interactions with UNC-49B, or by sterically interfering with UNC-49B lateral diffusion (*Santamaria et al., 2010*). Because NLG-1 stabilizes both forms of synaptic receptors, the contribution of the NLG-1 pool to post-synaptic currents could be mediated by immobilized or diffusing UNC-49 receptors (or a mixture of the two).

## Both scaffolds increase post-synaptic noise

Synaptic responses are intrinsically noisy, varying considerably even among inputs to the same cell. This noise arises from variability in both pre- and post-synaptic processes (*Lisman et al., 2007*). Prior studies suggested that an important contributor to post-synaptic noise is variation in receptor levels between synapses (*Nusser et al., 1997*), which can be adjusted bidirectionally by activity (*Bannai et al., 2009*; *Petrini et al., 2014*).

How do FRM-3 and NLG-1 alter quantal size? Prior modelling studies suggest that the non-homogeneous distribution of synaptic receptors into nanoclusters has profound effects on synaptic transmission (*MacGillavry et al., 2013*; *Nair et al., 2013*). Although these studies analyzed glutamatergic synapses, in the following we assume that similar principles apply to GABAergic synapses. Within post-synaptic elements, ~60% of receptors are localized in immobile nanoclusters (mean diameter 75 nm, mean receptor number 25) while the remaining ~40% are mobile and diffusely distributed (*MacGillavry et al., 2013*; *Nair et al., 2013*). The effect of receptor nanoclusters on transmission results from the fact that a single synaptic vesicle (SV) fusion activates receptors in a subsynaptic domain. For glutamate, the domain of activated receptors is estimated to have diameter ~200 nm (*Raghavachari and Lisman, 2004*). The size of the GABA domain has not been calculated, but is likely to be larger (due to the higher affinity of GABA$_A$ receptors). SV fusions at *C. elegans* cholinergic and GABAergic NMJs occur in an active zone with a diameter of 700 nm (*Hammarlund et al., 2007*; *Watanabe et al., 2013*). Thus, the spatial extent of GABA in the synaptic cleft will vary depending on the location of the SV fusion event within the active zone (*Barberis et al., 2011*). Consequently, quantal amplitude will vary depending on the proximity of the vesicle fusion site to the receptor nanocluster and the density of receptors in each nanocluster (*Franks et al., 2003*; *MacGillavry et al., 2013*; *Nair et al., 2013*). By contrast, mobile (unclustered) receptors have lower but uniform surface density; consequently, mobile receptors are predicted to mediate smaller quantal responses that have lower CV (*MacGillavry et al., 2013*). Our results are largely consistent with these modelling studies. The FRM-3 and NLG-1 scaffolds increase mIPSC amplitude and CV, presumably due to an increase in the number of UNC-49 nanoclusters at synapses or an increase in the number of receptors in each nanocluster.

## Implications for understanding psychiatric disorders

Changes in synaptic inhibition are proposed to play an important role in the pathophysiology of several neuropsychiatric disorders. Decreased inhibition is implicated in autism spectrum disorders (ASD) (*Rubenstein and Merzenich, 2003*), whereas excess inhibition has been proposed to occur in mental retardation syndromes, such as Down and Rett Syndromes (*Kleschevnikov et al., 2004*; *Dani et al., 2005*). Recurrent mutations in Neurexin, Neuroligin, and CASK are found in ASD (*Sanders et al., 2012*; *O'Roak et al., 2012*). We propose that these mutations may directly alter inhibitory transmission by altering the synaptic confinement of GABA$_A$ receptors. These results provide additional biochemical links between ASD associated genes and inhibitory transmission. As all of these molecules are conserved, the mechanisms we describe for confining synaptic GABA$_A$ receptors are likely to be conserved in other systems, including humans.

# Materials and methods

## *C. elegans* strains and mutant alleles

Strains were maintained at 20˚C under standard conditions. OP50 *Escherichia coli* was used as a food source for all experiments except electrophysiology where HB101 *E. coli* was utilized. A description of all alleles can be found at www.wormbase.org. The following strains were utilized in this study:

KP5330 nlg-1(ok259)
KP7320 nrx-1(nu485)
KP7338 frm-3(gk585)
CB1309 lin-2(e1309)
CB407 unc-49(e407)
MT106 lin-7(n106)
KP7637 lin-10(n1508)
KP7532 nrx-1(nu485); nlg-1(ok259)
KP7367 frm-3(gk585) nlg-1(ok259)
KP7534 nrx-1(nu485); frm-3(gk585)
KP7514 frm-3(gk585) lin-2(e1309)
KP5931 nuIs283 [Pmyo-3::unc-49::gfp::unc-54 3′UTR; Punc-25::snb-1::mcherry::unc-54 3′UTR]
KP7341 nuIs283; frm-3(gk585)
KP7478 nuIs283; nrx-1(nu485)
KP7133 nuIs283; lin-2(e1309)
KP7596 nuIs283; nlg-1(ok259)
KP7474 nuIs283; frm-3(gk585) nlg-1(ok259)
KP7340 nuIs283; frm-3(gk585) lin-2(e1309)
KP7597 nuIs283; unc-49(e407);
KP7545 nuIs522 [Pmyo-3::lin-2::mcherry::unc-54 3′UTR];
KP7552 nuIs523 [Pmyo-3::pHluorin::unc-49::unc-54 3′UTR];
KP7614 nuIs523; nlg-1(ok259);
KP7615 nuIs523; frm-3(gk585);
KP7553 nuIs524 [Pmyo-3::gfp::frm-3::unc-54 3′UTR];
KP7364 nuEx490 [Pfrm-3::gfp::unc-54 3′UTR];
KP7363 nuEx489 [Plin-2::gfp::unc-54 3′UTR];
KP7631 nuIs532 [Pmyo-3::NLG-1::mcherry::unc-54 3′UTR];
EN2630 LGII, krSi2 [Punc-49::unc-49B-RFP; unc-49(e407)
EN3224 LGII, krSi2 [Punc-49::unc-49B-RFP; unc-49(e407); nlg-1(ok259)
KP7894 LGII, krSi2 [Punc-49::unc-49B-RFP; unc-49(e407); frm-3(gk585)
KP7893 nuIs531 [Pmyo-3::frm-3::2flag::unc-54 3′UTR]; nuIs283

The *nrx-1(nu485)* null allele was isolated using the mosDEL protocol (*Frokjaer-Jensen et al., 2010*). The *C. elegans* transposon insertion line ttTi26330 was obtained from the NemaGENETAG consortium. 10.416 kb of the *nrx-1* locus was replaced with a Cb-unc-119(+) selectable marker (*Frokjaer-Jensen et al., 2010*). The engineered deletion included 982bp of upstream sequence and the first 9434bp of reference transcript C29A12.4a. The *nu485* mutation deletes exons 1–20, or 85%, of the *nrx-1* coding sequence. Exons 21–27, downstream of the ttTi26330 insertion site, were left intact. These could potentially encode a 234-residue C-terminal NRX-1 fragment. The EN2630 and EN3224 strains were kindly provided by Dr. Jean-Louis Bessereau.

## Plasmids

All expression vectors are based on the pPD49.26 backbone (A. Fire). Standard methods were utilized to construct all plasmids. A 3 kb *myo-3* myosin promoter was used for expression in body muscles, a 1.2 kb *unc-25* GAD promoter was used for expression in GABAergic neurons. The transcriptional reporter for *frm-3* and *lin-2* used 5 kb of 5′ flanking sequence. *nlg-1* (C40C9.5c), *nrx-1α* (C29A12.4a), *nrx-1β* (C29A12.4f), *frm-3* (H05G16.1), *lin-2* (F17E5.1a), and *unc-49B* (T21C12.1b) cDNAs were cloned from a cDNA library using primers corresponding to the predicted start and stop codons of the indicated EST. The ERM domain of *frm-1*, *frm-2*, *frm-3*, *frm-4*, *frm-5.1*, *frm-8*, *frm-9*, *frm-10*, *max-1*, *kin-32*, *kri-1*, *erm-1*, *nfm-1* cDNAs were cloned from a cDNA library using primers corresponding to the predicted start and stop codons of ERM domain. *nuIs524* contains GFP tagged FRM-3 cDNA constructs in which GFP was inserted in frame in the C-terminus. *nuIs523* contains pHluorin tagged UNC-49B cDNA constructs in which super-ecliptic pHluorin was inserted in-frame immediately after the signal peptide sequence. *nuIs522* contains an mCherry-tagged LIN-2 cDNA construct, in which mCherry is inserted in the N-terminus. *nuIs532* contains an mCherry-tagged NLG-1 cDNA construct, in which mCherry was inserted 13 residues from the carboxy-terminus (leaving the predicted PDZ ligand

intact). *nuIs531* contains an FLAG-tagged FRM-3 cDNA construct, containing two copies of the FLAG epitope at the C-terminus. Full descriptions of all plasmids are available upon request. Transgenic animals were prepared by microinjection, and integrated transgenes were isolated following UV irradiation.

## Yeast-two-hybrid

For yeast two-hybrid screens all the ERM domain containing proteins, Y2HGold yeast cells were co-transformed with pGADT7-UNC-49-TM3-4 loop and all the pGBKT7-ERM constructs respectively (BD clontech). Transformants selected from the SD-Leu-Trp plates were restreaked onto SD-Leu-Trp-His-Ade plates to test interactions. False positive from autoactivation was ruled out by co-transformation of pGBKT7-ERM construct with pGADT7 empty vector alone.

## Electrophysiology

Strains for electrophysiology were maintained at 20℃ on plates seeded with HB101. Cyanoacrylate glue was used to immobilize adult worms on the Sylgard-coated coverslip. The dissected adult worms were superfused in an extracellular solution containing 127 mM NaCl, 5 mM KCl, 26 mM $NaHCO_3$, 1.25 mM $NaH_2PO_4$, 20 mM glucose, 1 mM $CaCl_2$, and 4 mM $MgCl_2$, bubbled with 5% $CO_2$, 95% $O_2$ at 20℃. Endogenous GABA IPSC recordings were carried out at 0 mV using an internal solution containing 120 mM $CsCH_3SO_3$, 4 mM CsCl, 15 mM CsF, 4 mM $MgCl_2$, 5 mM EGTA, 0.25 mM $CaCl_2$, 10 mM HEPES, and 4 mM $Na_2ATP$, adjusted to pH 7.2 using CsOH. For muscimol-activated current recordings, 100 μM muscimol was pressure ejected for 1.0 s onto body muscles of adult worms.

## Microscopy

For colocalization studies, images were captured using a 60x objective (NA 1.45) on an Olympus FV-1000 confocal microscope at 5× digital zoom. Worms were immobilized with 30 mg/ml 2,3-Butanedione monoxamine (Sigma, St. Louis, MO, United States). UNC-49B puncta fluorescence was quantified by wide field fluorescence microscopy in, the dorsal nerve cord of young adults (midway between the vulva and the tail). Images were acquired using a Zeiss Axioskop I, Olympus PlanAPO 100× 1.4 NA objective, and a CoolSnap HQ CCD camera (Roper Scientific, Tuscon, AZ, United States). Maximum intensity projections of Z-series stacks were made using Metamorph 7.1 software (Molecular Devices, Sunnyvale, CA, United States). Line scans of dorsal cord fluorescence were analyzed in Igor Pro (WaveMetrics, Lake Oswego, OR, United States) using custom-written software. Mean fluorescence of 0.5 μm FluoSphere beads (Thermo Fisher, Waltham, MA, United States), which was measured during each experiment, was used to control the illumination intensity. All fluorescence values are normalized to wild type controls to facilitate comparison. All p-values indicated were based on ONE-way ANOVA or student t-tests.

## Fluorescence recovery after photobleaching (FRAP)

For FRAP studies, images were captured using a 60x objective (NA 1.45) on an Olympus FV-1000 confocal microscope at 5× digital zoom. Worms were immobilized with 0.1 μm polystyrene microspheres (Polysciences), and pads composed of 10% agarose in M9. To image GFP-UNC-49B, pH-UNC-49B, and RAB-3-mCherry, we used 0.5% power from a 473 nm (GFP) and 559 nm (mCherry) solid state diode laser. Five frames of GFP and mCherry signals were recorded prior to photobleaching. Photobleaching was achieved by one scan (90% power from the 473 nm laser) on a square region of interest (ROI, about 1.5 × 1.5 μm) that covered a single GFP punctum. GFP and mCherry signals were further recorded for 5 min at 0.2 frames/second. To eliminate motion artifacts, traces with more than 10% changes in mCherry signals were discarded. UNC-49B mobile fractions were calculated in MATLAB by fitting the data with a single exponential function, $I_{frap}(t) = A(1-e^{-\tau t})$, where A is reported as the mobile fraction. Statistical significance was determined using ONE-way ANOVA or Student's t test and all values reported are means ±SEM.

## Co-Immunoprecipitation

Extracts were prepared from mixed staged worms expressing GFP-UNC-49 or GFP-UNC-49 and FLAG-FRM3 using a microfluidizer in buffer A (50 mM HEPES PH7.7, 50 mM KAc, $2mMgAc_2$, 250 mM sucrose, 1 mM EDTA and proteinase inhibitors). Worm extracts were clarified by centrifugation (15 min, 7000g). Membranes were isolated from the resulting supernatant by high speed centrifugation (40 min, 45,000g). Membrane proteins were solubilized with a dounce homogenizer

(5 times) in IP buffer (20 mM HEPES PH7.4, 150 mM NaCl, 2 mM MgCl$_2$, 0,1 mM EDTA, 1% Triton and proteinase inhibitors), and incubated with FLAG M2 affinity gel (A2220) overnight at 4°. Flag affinity gel were washed three times with IP buffer, and eluted with loading buffer for Western blot.

## Acknowledgements

We thank the following for strains and reagents: the *C. elegans* Genetics Stock Center and Jean-Louis Bessereau. We thank members of the Kaplan lab, Peter Juo, Jihong Bai, and Jeremy Dittman for helpful discussions and critical comments on the manuscript.

## Additional information

### Funding

| Funder | Grant reference | Author |
| --- | --- | --- |
| National Institutes of Health (NIH) | R01 NS32196 | Xia-Jing Tong, Zhitao Hu, Dorian Anderson, Joshua M Kaplan |
| Simons Foundation | SF273555 | Xia-Jing Tong, Zhitao Hu, Yu Liu, Dorian Anderson, Joshua M Kaplan |

The funders had no role in study design, data collection and interpretation, or the decision to submit the work for publication.

### Author contributions

X-JT, ZH, Conception and design, Acquisition of data, Analysis and interpretation of data, Drafting or revising the article; YL, Acquisition of data, Analysis and interpretation of data, Drafting or revising the article; DA, Conception and design, Contributed unpublished essential data or reagents; JMK, Conception and design, Analysis and interpretation of data, Drafting or revising the article

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
