## [Decision Letter]

Thank you for submitting your work entitled “A network of Autism linked genes stabilizes two pools of synaptic GABA_A_ receptors” for peer review at *eLife*. Your submission has been favorably evaluated by Gary Westbrook (Senior Editor) and three reviewers, one of whom is a member of our Board of Reviewing Editors. One of the three reviewers, Kang Shen (Reviewer 2), has agreed to reveal his identity.

The reviewers have discussed the reviews with one another and the Reviewing Editor has drafted this decision to help you prepare a revised submission.

Previous single molecule-receptor tracking experiments have identified mobile and immobile pools of postsynaptic receptors at excitatory and inhibitory synapses, although these results rely on mostly in vitro results. Tong et al. employ in vivo approaches to demonstrate that there are mobile and immobile pools of GABA_A_ receptors (UNC-49) at *C. elegans* neuromuscular junctions. In addition, they explore molecular mechanisms that control UNC-49 localization and report distinct phenotypes for two different scaffolds, one consisting of the trans-synaptic neurexin-neuroligin complex and one consisting of a cytoplasmic scaffold containing FRM-3 and LIN-2. Mutation of each scaffold individually produces specific phenotypes with respect to receptor mobility (assessed by FRAP) and synaptic transmission. Importantly, the authors find that FRM-3 and NLG-1, function in parallel to recruit GABA_A_ receptors to synapses, and that simultaneous inactivation of both scaffolds results in a near complete loss of GABAergic transmission and receptor stabilization at synapses.

This work complements two very recent studies (Shen and Bessereau labs, Neuron, 2015) that reported the importance of NLG-1 in synaptic recruitment of UNC-49. The identification of a function for FRM-3 for synaptic UNC-49 concentration is novel and important. Moreover, an important concept advanced by this work is the presence of multiple systems that act in parallel to stabilize functionally different pools of postsynaptic neurotransmitter receptors at the same class of synapse. This concept is backed up by strong in vivo data, and is accompanied by specific molecular mechanisms that stabilize the two different pools of receptors.

Essential revisions:

1) The authors show that the UNC-49 fluorescence is reduced in the *nlg-1* mutant. They also show that the FRAP rate is modified by the *nlg-1* mutation and that the *frm-3* mutant phenotype is greatly enhanced by the *nlg-1* mutation. These data argue strongly that *nlg-1* and *frm-3* function in parallel, both contributing to the synaptic targeting of UNC-49. The one piece of data that does not support this model is the lack of changes of either mIPSC frequency or amplitude. Nevertheless, two recent papers (Tu et al. and Maro et al.) did report electrophysiological phenotypes in *nlg-1* mutant. Both papers showed a strong reduction of mIPSC frequency and a subtle reduction of mIPSC amplitude. All these results seem to be consistent with the parallel model. Based on the lack of observed functional phenotypes, the authors propose that the *nlg-1* anchored UNC-49 receptors do not contribute to the GABAergic synaptic transmission. The authors have several options to deal with this discrepancy. First, the authors can provide reasons why the recording conditions that they used are better than the two published reports. Or, they can provide additional evidence to show that the NLG-1-dependent UNC-49 receptors are inactive and do not contribute to synaptic currents. Or, the authors can modify their conclusions.

2) The data from the different genetic manipulations strongly suggests that there are different pools of UNC-49 defined by their differentially dependence on NLG-1 and FRM-3. What is not entirely clear is whether “mobile” and “immobile” is an accurate classification. If the NLG-1-dependent pool is “mobile” then why does mobility in *frm-3*, *nlg-1* double mutants increase compared to *frm-3* mutants alone? The authors suggest that this “immobilization of the mobile pool” is correlated to these receptors participating in synaptic transmission in the mutant context but a mechanism to explain this is not offered. Additional data or indications to clarify this point this would greatly improve the paper.

3) One potential weakness of the receptor mobility (FRAP) experiments is the use of transgenically overexpressed UNC-49 receptors. Thus, the size of mobile and immobile pools may differ between conditions. What supports the validity of the transgenically expressed receptors is that the mIPSC amplitudes and rates are similar to wild-type. Also, the magnitude of loss of UNC-49-GFP fluorescence in *nlg-1* mutants is similar to what was reported by Bessereau and colleagues who used a tag introduced into the endogenous *nlg-1*. As the mobility measurements are a central part of this study this issue needs to be addressed somewhere in the text.

4) This study does not detect a significant reduction in mIPSC amplitude in *nlg-1* mutants, which is different from the Bessereau and Shen work. In the Discussion the authors point out that *nlg-1* mutant animals are behaviorally normal and suggest that this supports their observation that GABAergic transmission is not severely impaired. This important point could be made much more clearly if the authors included data on this observation for *nlg-1*, *frm-3* and *nlg-1:frm-3* double mutants.

5) The authors state in Discussion that “mobile NLG-1-stabilized UNC-49 receptors can be actively recruited to the receptor field that mediates synaptic responses, and that this recruitment is inhibited by pre-synaptic NRX-1α”. Given that NLG-1 mutation suppresses UNC-49B puncta (Figure 9C), perhaps NRX-1α is inhibiting the translocation of NLG-stabilized UNC-49B from a subsynaptic region that is not affected by GABA to an adjacent subsynaptic region that mediates GABA synaptic transmission in the same synapse? If this is the case, how is it possible that GABA released from the nerve terminal acts so specifically on a subsynaptic region but not on another in the same inhibitory synapse? This should be clarified.

---

## [Author Response]

Essential revisions:

*1) The authors show that the UNC-49 fluorescence is reduced in the* nlg-1 *mutant. They also show that the FRAP rate is modified by the* nlg-1 *mutation and that the* frm-3 *mutant phenotype is greatly enhanced by the* nlg-1 *mutation. These data argue strongly that* nlg-1 *and* frm-3 *function in parallel, both contributing to the synaptic targeting of UNC-49. The one piece of data that does not support this model is the lack of changes of either mIPSC frequency or amplitude. Nevertheless, two recent papers (Tu et al. and Maro et al.) did report electrophysiological phenotypes in* nlg-1 *mutant. Both papers showed a strong reduction of mIPSC frequency and a subtle reduction of mIPSC amplitude. All these results seem to be consistent with the parallel model. Based on the lack of observed functional phenotypes, the authors propose that the nlg-1 anchored UNC-49 receptors do not contribute to the GABAergic synaptic transmission. The authors have several options to deal with this discrepancy. First, the authors can provide reasons why the recording conditions that they used are better than the two published reports. Or, they can provide additional evidence to show that the NLG-1-dependent UNC-49 receptors are inactive and do not contribute to synaptic currents. Or, the authors can modify their conclusions.*

*4) This study does not detect a significant reduction in mIPSC amplitude in* nlg-1 *mutants, which is different from the Bessereau and Shen work. In the Discussion the authors point out that* nlg-1 *mutant animals are behaviorally normal and suggest that this supports their observation that GABAergic transmission is not severely impaired. This important point could be made much more clearly if the authors included data on this observation for* nlg-1*,* frm-3 *and* nlg-1:frm-3 *double mutants.*

5) The authors state in Discussion that “mobile NLG-1-stabilized UNC-49 receptors can be actively recruited to the receptor field that mediates synaptic responses, and that this recruitment is inhibited by pre-synaptic NRX-1α”. Given that NLG-1 mutation suppresses UNC-49B puncta (Figure 9C), perhaps NRX-1α is inhibiting the translocation of NLG-stabilized UNC-49B from a subsynaptic region that is not affected by GABA to an adjacent subsynaptic region that mediates GABA synaptic transmission in the same synapse? If this is the case, how is it possible that GABA released from the nerve terminal acts so specifically on a subsynaptic region but not on another in the same inhibitory synapse? This should be clarified.

These three comments all refer to the same issue, our failure to detect a significant mIPSC defect in *nlg-1* single mutants. Prompted by these comments (and by Tu and Maro’s results), we decided to repeat our analysis of *nlg-1* mutants. Our new experiments indicate that *nlg-1* mutants have significant reductions in mIPSC rate and amplitude, similar to results reported by Tu and Maro. We thank the reviewers for urging us to revisit the *nlg-1* phenotype in our revised manuscript. These new results are described in the revised text (please see the following passage in the Results: “The mIPSC rate was significantly reduced in *nlg-1* mutants […] the decreased mIPSC rate could result from decreased pre-synaptic GABA release”).

In the Discussion, we also add that inactivating NLG-1 caused corresponding decreases in total synaptic UNC-49B (30% decrease), mobile UNC-49B (40% decrease), and mIPSC amplitudes (48% decrease). Thus, analysis of single mutants suggests that FRM-3 and NLG-1 stabilized receptors contribute equally to post-synaptic currents.

We do not know why our prior analysis failed to detect the *nlg-1* mIPSC defect. The new results have been replicated multiple times (n=28), the *nlg-1* defect is fully rescued by a muscle transgene, and these results are more similar to those reported by Tu and Maro. For these reasons, we are very confident in our new conclusions. Given these new results, we decided to confirm mIPSC rate and amplitude phenotypes for several additional genotypes in our paper, including: *frm-3* (n=8), *nrx-1* (n=39), *frm-3 nlg-1* double mutant (n=10), and *nrx-1;nlg-1* double mutant (n=18). These new experiments replicated the results in our original submission. In the revised manuscript, the mIPSC data reported for *nlg-1* and *nrx-1; nlg-1* double mutants are new. The original data is reported for all other genotypes.

2) The data from the different genetic manipulations strongly suggests that there are different pools of UNC-49 defined by their differentially dependence on NLG-1 and FRM-3. What is not entirely clear is whether “mobile” and “immobile” is an accurate classification.

We agree that the NLG-1-stabilized pool comprises a mixture of both mobile and immobilized receptors. We attempt to explain this more clearly in our revised Discussion (please see: “Two results suggest that the NLG-1 stabilized pool […] could be mediated by immobilized or diffusing UNC-49 receptors (or a mixture of the two)”).

Despite this complexity, we believe it is accurate to state that NLG-1-stabilized synaptic receptors are more mobile than those stabilized by FRM-3.

*If the NLG-1-dependent pool is “mobile” then why does mobility in* frm-3*,* nlg-1 *double mutants increase compared to* frm-3 *mutants alone? The authors suggest that this “immobilization of the mobile pool” is correlated to these receptors participating in synaptic transmission in the mutant context but a mechanism to explain this is not offered. Additional data or indications to clarify this point this would greatly improve the paper.*

Thanks for pointing this out. Our original submission did not explain this result clearly. In FRAP experiments, we measure the ability of extra-synaptic (i.e. unbleached) receptors to restore synaptic fluorescence following photobleaching. These extra-synaptic receptors are mobile UNC-49 receptors on the cell surface, which we also detect in our muscimol-activated current recordings. In the *frm-3 nlg-1* double mutant, we argue that synaptic receptors are eliminated and that all remaining receptors are extra-synaptic receptors in the muscle plasma membrane. Prompted by this comment, we revised our discussion to explain this result more clearly (“Mobile receptors detected by FRAP in *frm-3 nlg-1* double mutants likely correspond to extra-synaptic receptors on the cell surface (which mediate muscimol-activated currents)”).

*3) One potential weakness of the receptor mobility (FRAP) experiments is the use of transgenically overexpressed UNC-49 receptors. Thus, the size of mobile and immobile pools may differ between conditions. What supports the validity of the transgenically expressed receptors is that the mIPSC amplitudes and rates are similar to wild-type. Also, the magnitude of loss of UNC-49-GFP fluorescence in* nlg-1 *mutants is similar to what was reported by Bessereau and colleagues who used a tag introduced into the endogenous* nlg-1*. As the mobility measurements are a central part of this study this issue needs to be addressed somewhere in the text.*

This is another good point. To address this issue, we repeated our FRAP experiments using the single copy RFP-tagged UNC-49B transgene described by Bessereau and colleagues. Please see the subsections “LIN-2A and FRM-3 stabilize immobile UNC-49B receptors at GABAergic NMJs” (“The GFP-UNC-49B transgene is likely to be expressed at higher levels […] a similar increase in FRAP of RFP-UNC-49B was observed in *frm-3* mutants (Figure 4—figure supplement 1)”) and “NLG-1 Neuroligin stabilizes mobile UNC-49B at synapses” (“FRAP of GFP-UNC-49B […] residual synaptic UNC-49B receptors were largely immobile”).